

# Biologically-oriented mud volcano database: muddy_db

Alexei Remizovschi and Rahela Carpa

Department of Molecular Biology and Biotechnology, Faculty of Biology and Geology, Babes-Bolyai University, Cluj-Napoca, Cluj, Romania

## ABSTRACT

Mud volcanoes (MVs) are naturally occurring hydrocarbon hotbeds with continuous methane discharge, contributing to global warming. They host microbial communities adapted to hydrocarbon oxidation. Given their research value, MVs still represent a niche topic in microbiology and are neglected by hydrocarbon-oriented research. All the data regarding MVs is sporadic and decentralized. To mitigate this problem, we built a custom Natural Language Processing pipeline (muddy_mine), and collected all the available MV data from open-access articles. Based on this data, we built the muddy_db database. The muddy_db represents the first biologically oriented database rendered as a user-friendly web app. This database includes all the relevant MV data, ranging from microbial taxonomy to hydrocarbon occurrence and geology. The muddy_mine and muddy_db tools are licensed under the GPLv3. muddy_db R Shiny web app: https://muddy-db.shinyapps.io/muddy_db/ muddy_db R package: https://github.com/TracyRage/muddy_db muddy_mine Conda package: https://github.com/TracyRage/muddy_mine.

Corresponding author
Rahela Carpa,
rahela.carpa@ubbcluj.ro

# INTRODUCTION

Mud volcanoes (MVs) represent hydrocarbon discharging landforms (*Mazzini & Etiope, 2017*). They are distributed worldwide in both marine and terrestrial environments (*Milkov, 2000*). The most distinctive feature of MVs is recurrent methane emission. Due to methane emissions, MVs contribute extensively to global warming (*Etiope, Feyzullayev & Baciu, 2009*).

MV genesis is mainly caused by a naturally mediated process - kerogen maturation (*Vandenbroucke & Largeau, 2007*). Therefore, the surrounding area of MVs can provide valuable data regarding both aerobic and anaerobic hydrocarbon microbial oxidation (*Cheng et al., 2012*).

Over the years, MV research has mainly focused on anaerobic oxidation of methane (AOM) and the implicit interaction between sulfate-reducing bacteria and methane oxidizing archaea (ANME) (*Bose et al., 2013*; *Cui et al., 2014*). In addition to AOM research, MVs were also investigated in the context of hydrocarbon research. A myriad of MV studies discussed the thermogenic and biogenic origin of evolved methane (*Etiope, Feyzullayev & Baciu, 2009*; *Sano et al., 2017*).
Despite these studies, the biological aspect of MVs is still a niche and unexplored topic. The biological data regarding MVs are sporadic and mostly biased towards AOM. Even worse, the already available data is not centralized. An MV-dedicated database would facilitate this, however, making it easier for researchers to conduct comparative studies.

Meanwhile, mainstream biomedical fields have extensively employed natural language processing (NLP) techniques to mine meaningful data from research articles (*Wang et al., 2020*). Simultaneously, the number of databases related to biomedical fields is considerable (*Luo et al., 2016*). Niche environmental science fields have not entirely caught up. This lack of tools limits the possibility to mine environmental-oriented articles and build field-specific databases, delaying the publication of the meta-analyses or any comparative studies.

Fortunately, democratic NLP models and tools have been published over the last years. Some of them can be easily used by environmental scientists with limited computer science (CS) experience, for example, the spaCy library, ScispaCy models, and S2ORC database (*Honnibal & Johnson, 2015*; *Neumann et al., 2019*; *Lo et al., 2020*).

These advancements in NLP provide opportunities for consolidating and promoting niche environmental topics such as MV microbiology.

In this paper, we used them to build the first biologically oriented mud volcano database, muddy_db. This niche database consolidates all relevant biological and environmental data, that will be of great use for researchers specializing in bacterial hydrocarbon oxidation or MV microbiology. Our pipeline can serve as a methodological blueprint for other research communities interested in employing NLP to build their specialized databases.

## METHODS

To collect all the available data regarding the biological aspects of MVs, we exclusively relied on open-access articles. This minimized potential issues arising from both copyright concerns and the lack of standardized XML encoding of open-access literature.

We then built a custom pipeline muddy_mine to extract all the biologically oriented tokens, including taxonomy-, chemicals-, geology-, and MV-specific terms (Fig. 1). Tables generated by muddy_mine are presented *via* the muddy_db framework (Fig. 2).

### Data collection

We used the S2ORC (20200705v1) database to collect open-access articles. S2ORC represents a centralized database that includes 12.7 million articles with a fully preserved paper structure. S2ORC is quite comprehensive and includes niche environment science articles (*Lo et al., 2020*). Given these facts, we extracted all the available MV-related titles ($N = 118$ total, $N = 115$ deduplicated) from the S2ORC.

### Token extraction and muddy_mine pipeline

Having MV articles, we proceeded with token extraction (*i.e.,* the extraction of the terms of interest) using the muddy_mine pipeline.

Taxonomy extraction represented a difficult challenge due to the fact that we intended to collect as many tokens as possible. To overcome this problem, we used the spaCy library

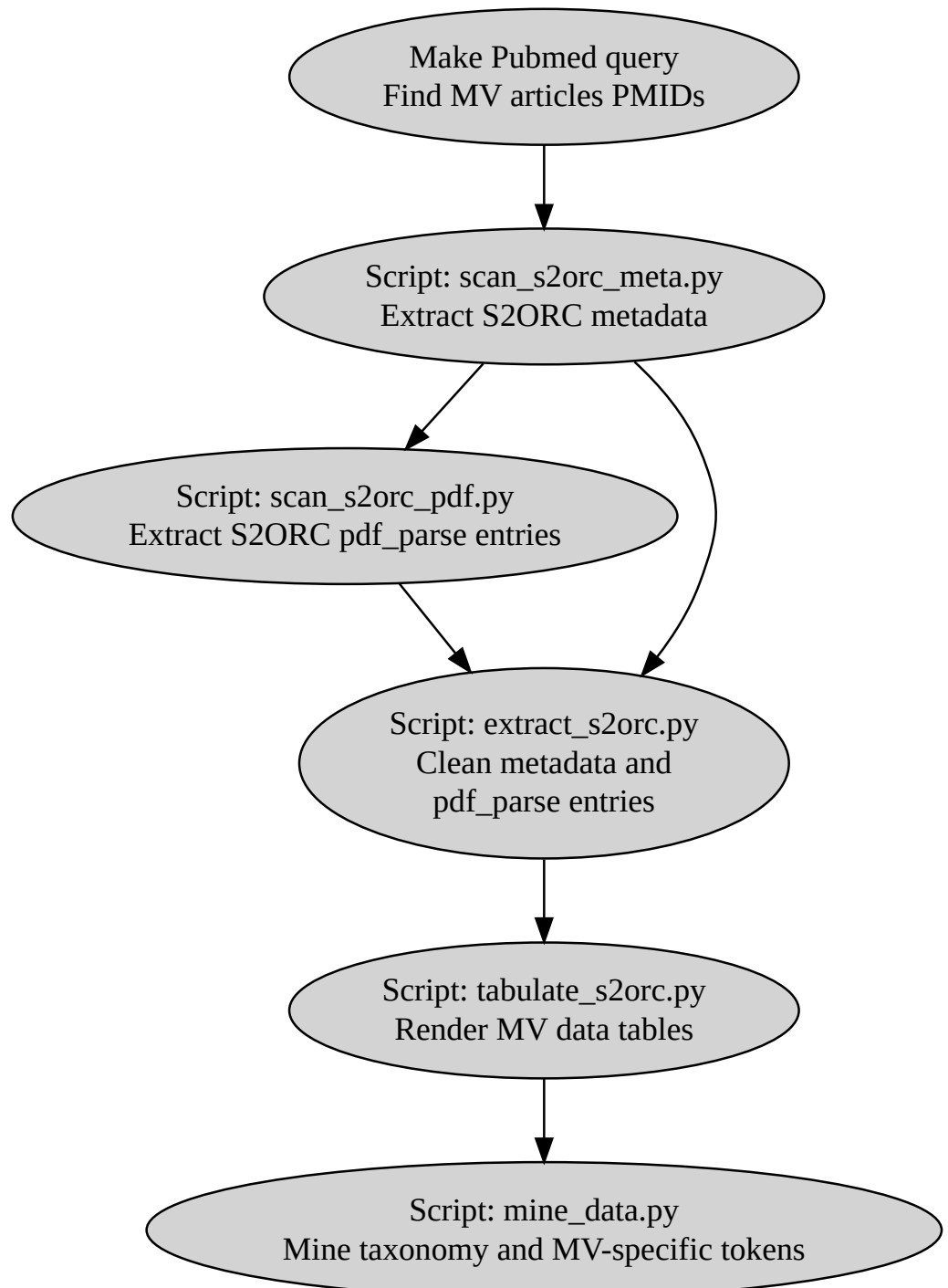

**Figure 1  muddy_mine-pipeline used to build the muddy_db database.** MV, mud volcano; PMIDs, Pubmed IDs.

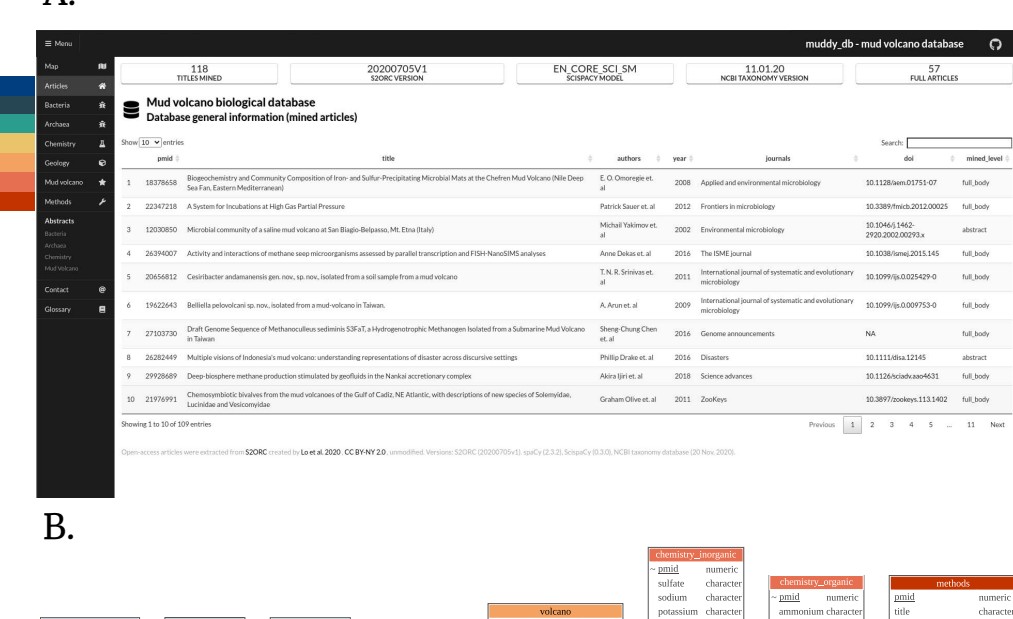

**Figure 2** **(A) muddy_db general appearance. (B) muddy_db schema.** For each muddy_db tab, there is a reciprocal table in the database (colorwise). Pubmed ID (PMID), database primary key.

(2.3.2), third-party ScispaCy NLP models (0.3.0), and the most recent NCBI Taxonomy database (20 November, 2020) (*Honnibal & Johnson, 2015*; *Neumann et al., 2019*; *Schoch et al., 2020*). First, we extracted all the taxon tokens using en_core_sci_sm ScispaCy model (2.2.5). Second, we checked those tokens against a local NCBI Taxonomy database. Third, we counted the extracted tokens. The more often a token occurs, the counting number, the more likely the token was explicitly discussed in the article.

By iterating this three-step algorithm (each iteration being focused on a specific taxonomic rank), we managed to centralize MV-specific taxonomy on all the possible levels: phylum, class, order, family, and genus.

The same procedure was applied to mine other types of information. We extracted and counted the tokens related to the following categories: chemistry (inorganic ions, hydrocarbons), geology (geological periods, minerals), MV terminology (ANME, methanogenesis type), and experimental methods (PCR types, amplified genes, chromatography). The full list can be found in the muddy_db repository.

The raw output of the muddy_mine pipeline is a set of csv tables with MV data.

### Building muddy_db database

After obtaining muddy_mine raw output, we can advance to the next step - building a web based database interface to present the output in a user-friendly manner. To this end, we created a Shiny web app, entitled muddy_db. In order to build it, we used the following R packages: shiny (1.5.0), semantic.dashboard (0.2.0), and golem (0.2.1) (*Filip & Igras, 2021*; *Chang et al., 2021*; *Fay et al., 2021*). This app includes all the output generated by the muddy_mine pipeline. Specifically, it displays tokens and their counts, extracted both from the article bodies ($N = 57$) and abstracts ($N = 115$). Additionally, we added annotated map, which displays the geographical distribution of MVs and their affiliated research metadata. To build it, we used leaflet package (2.0.4.1) (*Cheng, Karambelkar & Xie, 2021*).

### System requirements

The muddy_mine pipeline was designed to run on systems with modest memory requirements. We achieved this feature by using Python generators (*Van Rossum & Drake, 2009*). Intel Core i3 (3rd Gen) 3217U / 1.8 GHz processor (Intel, USA) and 4GB RAM system was used to build muddy_db. The mining process took 24 h.

## RESULTS

The aim of the muddy_db is to gather all the available MV biologically relevant data and include it in a user-friendly database (Fig. 2). First, we collected all the known taxa associated with MVs. The muddy_db includes data regarding archaeal and bacterial taxonomy on all the possible taxonomy levels. Second, we gathered information regarding metabolic pathways, geology, and chemical substrate availability. Together, these conveniently centralized taxonomic and physicochemical data can facilitate routine MV-related documentation. Additionally, muddy_db contains data regarding experimental methods, applied in the context of MV studies, which can guide specialists to implement appropriate research strategies.

## DISCUSSION

MVs are considered to be one of the settings where early life evolved (*Pons et al., 2011*). They sustain a plethora of bacterial metabolic pathways, ranging from methane oxidation and synthesis to sulphate reduction (*Kleindienst et al., 2014*; *Cheng et al., 2012*). These pathways and their affiliated microbial communities could provide valuable data regarding(1) origin of microbial life, (2) the effect of the naturally occurring methane discharging systems on global warming, (3) the contribution of microbial consortia to oil souring (*Gieg, Jack & Foght, 2011*; *Etiope, Feyzullayev & Baciu, 2009*; *Pons et al., 2011*). The accumulation of MVs data could enhance knowledge regarding topics that range from fundamental studies to ecology and engineering.

Given these facts, MVs should be the main focus of hydrocarbon-oriented research and ecology. Unfortunately, data regarding the biological aspects of MVs are scarce. Additionally, the data already gathered are not combined in a dedicated database. The creation of a specialized MV database will help future research efforts, and thus help raise the profile of MV microbiology research in environmental science

Biomedical fields have always represented the cutting-edge subset of natural science, which actively implement CS techniques, and are tightly intertwined with big data (*Luo et al., 2016*). Simultaneously, the implementation of CS methods in niche environmental fields lags. With the muddy_mine NLP pipeline and muddy_db database, we have both advanced the application of CS methods in environmental contexts and addressed a critical lack in the toolbox of MV microbiologists.

The creation of muddy_db tools aims to create a platform, that would provide sufficient data to perform meta-analyses or comparative studies of the MVs. We hope that muddy_db would facilitate the discovery of atypical taxonomic patterns and point out the influence of chemical substrate and geography on MV microbial characteristics. For example, muddy_db offers data regarding MV essential dichotomies such as the predominance of biogenic/thermogenic methane, presence of organic acids/hydrocarbons, and terrestrial/marine localization. All of these conjugated parameters influence the microbial distribution and metabolic patterns in MV sediments (*Bhattarai, Cassarini & Lens, 2019*; *Lazar et al., 2012*; *Wrede et al., 2012*; *Remizovschi et al., 2020*; *Sano et al., 2017*). Simultaneously, muddy_mine represents a reproducible example of a mining technique applied in the context of environmental studies. The muddy_mine pipeline, however, employs general NLP tools, and so could be used by researchers to mine their own data of interest.

Whilst we have shown muddy_mine and muddy_db to be effective, however, there are some caveats. We designed muddy_mine around the S2ORC repository, which provides a standardized encoding of open access literature. Additional parsers would be needed to ingest other article formats, such as TEI-XML and JATS-XML (*J4R, 2021*; *TEI Consortium, 2021*). Our reliance on S2ORC also ensures that the derived resource, muddy_mine, can be publicly shared without infringing author or journal copyright. This may not be the case for article collections obtained from other sources.

## CONCLUSIONS

The muddy_db represents the first biologically oriented mud volcano database. It was designed to provide a comprehensive data corpus that can facilitate mud volcano research and shed light on the topic as a whole. The muddy_db contains data ranging from taxonomy to geology and experimental methods. Simultaneously, the muddy_mine NLP pipeline can serve as an example of accessible implementation of NLP techniques in environmental sciences.

### Funding
The authors received no funding for this work.

### Competing Interests
The authors declare there are no competing interests.

## Author Contributions

- Alexei Remizovschi conceived and designed the experiments, performed the experiments, analyzed the data, prepared figures and/or tables, authored or reviewed drafts of the paper, and approved the final draft.
- Rahela Carpa conceived and designed the experiments, performed the experiments, authored or reviewed drafts of the paper, and approved the final draft.

## Data Availability

The muddy_db R Shiny web app is available at: https://muddy-db.shinyapps.io/muddy_db/

The muddy_db R package is available at GitHub: https://github.com/TracyRage/muddy_db

The muddy_mine Conda package is available at GitHub: https://github.com/TracyRage/muddy_mine.

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
