# Peer review of "Biologically-oriented mud volcano database: muddy_db"

_PeerJ, doi:10.7717/peerj.12463_

## Round 0.1 · original submission · Major Revisions

Two reviewers have examined your paper. R1 in particular remarks on the timely nature and utility of the work. Both reviewers, however, note a number of areas where the manuscript can be improved to better communicate both the microbiological, ecological, hydrogeological, and computational aspects of the work. They also note that the reliance solely on open access literature may be a limiting factor for the database, and recommend the system be expanded to allow ingestion of literature from other sources.

In addition to addressing the reviewer's comments, I offer my own remarks below - the majority of the comments concern the manuscript itself rather than the work that is described - I hope that you find them useful when preparing your revised submission.

1. Overall style and presentation.
i. The introduction of the manuscript seem to consist of one sentence paragraphs - which makes it difficult to read and perhaps fool the reader into thinking the manuscript is 'unfinished'.
ii. The results section presents an 'informal schema' for the database. Surely this is better presented as a table (actually based on the 'schema' rather than inspired by it!) with detailed noted regarding the kinds of values observed in each column.
iii. Screenshots of the Shiny app in action would be a very effective way of demonstrating the utility of the system, and help to describe them better to the reader.

2. Reproducibility, and Reporting of methodology implemented in the pipeline and webapp. I was impressed by the comprehensive documentation accompanying the github hosted conda and R packages. However, I found that specific parameters used for muddy_db construction, and the processes involved in both literature mining and construction of the shiny app were not well described in the manuscript.
i. Please report the query used to construct muddy_db, and ideally also provide discussion explaining whether the query could be further improved. I found 'mud[TIAB] AND volcano[TIAB]' in one of the images of the README - but this is not sufficient.
ii. Please cite versions (or alternatively, download date) for the version of S2ORC used. Ideally dates and versions of the corpus used to build muddy_db should be also communicated in the Shiny app so users can communicate these should they employ muddy_db in their own work.
iii. Mention is made in the pipeline's documentation that mining can 'take some time' - and a reduced corpus is provided to test the pipeline - in the paper you should report timings and CPU/memory requirements for construction of the production version of muddy_db described in the paper.

3. Data processing.
i. https://github.com/TracyRage/muddy_db/blob/main/inst/article/methods.csv shows the methods extracted from papers found in muddy_db. In several cases, methods appear multiple times with different cases, or different bracketed numbers (e.g. 'salinity' and 'Salinity', '16S' and '16s'). Surely these should be normalised and grouped ? Also, what do the numbers actually mean ??
ii. Ideally, canonical definitions should be provided for all terms appearing in muddy_db fields - whilst such terms are of course familiar to experts, muddy_db will gain traction faster if it can also help people to learn about the terms that appear in the literature.

4. Threats to validity. You note that taxon extraction was challenging. Is there the possibility that the NLP based taxon recognition will not have 100% recall, particularly in the case of newly minted taxons ? Was this aspect tested during development ? It would help if you clarify what happens in this situation in your revised manuscript, and describe any procedures you used to validate the NLP pipeline. Whilst muddy_db is certainly useful, it is important to know if there are details of papers that may be missed due to limitations of its literature mining methodology.

5. Ingestion of supplementary materials for publications. It is likely that many publications of this type will involve supplementary data hosted at the journal or deposited in public archives - these are not presently included in S2ORC, but is it potentially feasible to incorporate these data into the database, and are there perhaps plans to do so in the future ?

Reviewer 1 ·

Basic reporting

The paper describes the establishment of a database that will aid researchers working on the microbiology and geochemistry of mud volcanoes. I applaud the authors for compiling this database. I only have a few suggestions:

- Line 11: In addition to bacteria, these systems also host archaea. I would also suggest to stay general and say that they host various bacteria and archaea after than consortia. Not all live as consortia.

- Line 30/31: change 'anthropic' to 'anthropogenic' . Also, I would suggest to rephrase the whole sentence as it is not clear what is meant.

- Line 50: I am not sure what is meant by 'biologically flavoured tokens'. Please reconsider rewriting by using different wording.

-Line 59: Again, I am not sure what is meant by 'Token extraction' and 'token' throughout paragraph. Please consider using different wording.

-Line 103: Consider changing to 'one of the settings where life evolved'

- For the web app, I would suggest to make it possible to include a link to the respective papers and data. That will make it easier for people to get the information.

- I understand that the authors were only including only data from open-access articles. However, it would be useful if the authors would include an option to allow investigators to add data from papers that were published in non-open-access articles.

Experimental design

no comment

Validity of the findings

no comment

Reviewer 2 ·

Basic reporting

This paper described the establishment of a database for mud volcanoes focusing on the biological aspects. Overall, the manuscript is clear and straightforward. However, the authors need to tone down the language in some parts as listed below. Also I suggest the authors to discuss more about why we should care about mud volcanoes. Some detailed suggestions are as follows.

L11 bacterial consortia -> microbial consortia, as both bacteria and archaea can oxidize hydrocarbons.
L13 NLP pipeline -> full name
L17 muddy db is indefinitely available to everyone. -> most likely not all.
L30-31 Pristine oxidation is not influenced by anthropic factors. -> what does this mean?
L28-L31 the authors need to provide more detailed background about MVs.
L36 the authors should explain why NLP is helpful for MV studies.
L45 again, not all the microbiology researchers
L51 had to build -> built
L63 Please also consider GTDB, GENOME TAXONOMY DATABASE; currently more people use GTDB.
L78 vs L58 Please discuss why N is different: 118 vs 115
L79 Rephrase, as it is most likely not indefinitely available for everyone
L99 Please explain DAMO

Experimental design

My main concern on the experimental design part is that the authors need to gather more papers to make the database much more comprehensive. The number of 57 is a rather low number. E.g. L49 I suggest, the authors also explore papers that are also available in fulltext in e.g. researchgate to increase the numbers

Validity of the findings

For this part, I suggest the authors to make DISCUSSION part more meaningful rather than repeat sentences from the introduction and results. E.g. the authors need to discuss both the merits and also the weaking points of the database.

---

## Round 0.2 · Minor Revisions

Thank you for the revised manuscript, and responding to reviewers' concerns from the previous round.

The new version of the manuscript is much improved, and it is my opinion that it is nearly ready for publication, providing several minor issues of clarity and reporting are addressed. To this end I have provided an annotated PDF containing suggested rewordings and highlighting points that require references or clarification.

I have 3 significant concerns.

1. I reject your explanation that 'normalisation' is not required because the variation in extracted terms provides 'context'. One of the most important things that a database allows (regardless of any particular design philosophy) is to reduce the human effort needed to relate identical observed datums across sets of observed data. Muddy_db's sequencing field includes both '16S' and '16s' - which clearly refer to the same methodology. If there is utility in preserving the token as it appeared, as you claim in your rebuttal, this should be done in a dedicated context view - e.g. as a tool tip shown when the mouse hovers over a token instance which shows the term as it appeared in the mined text.

I recommend you highlight these details as potential areas of improvement that might be addressed as further work, or through open source community development. It would be trivial, for instance, to regularise terms like the example above via controlled vocabularies such as those available from the EBI ontology look up service: https://www.ebi.ac.uk/ols/search?q=16S

2. Figures 2 and 3 lack legends to explain what is shown. In particular, Figure 2 shows a screenshot of the tool but explains nothing about what is shown, or how what is shown relate ot the database schema pictorially represented in Figure 3.

I recommend these two figures are merged into a single - two panel figure that shows the 'muddy_mine_db web app' and 'muddy_mine_db schema'. The legend should then communicate clearly how elements of the schema manifest in the web app.


3. Demonstrating that the application provides insight.

One of the most important roles of publications that accompany researcher-oriented applications and tools like muddy_mine's web app is to demonstrate that the tool can provide real insight for its users. In particular, you stated in lines 124-127 that:
"The creation of muddy db tools aims to create a platform, that would provide sufficient data to perform meta-analyses or comparative studies of the MVs. Specifically, we hope that muddy db would facilitate the discovery of atypical taxonomic patterns and point out the influence of geography on MVs characteristics."

Whilst the description of the web interface is accurate, a figure showing how the different characteristics of MV can be related to their geographic regions along with accompanying legend would be extremely valuable. Are there examples of such correlations already published that can be used to demonstrate how the webapp allows similar insights to be obtained ?

---

## Round 0.3 · accepted · Accept

Thank you for your revised manuscript, and responding to my requests for revision. It is a shame that you were not able to include additional examples demonstrating the potential insights from the muddy_db interface, but I realise that such examples are not easy to produce without conducting in-depth research. Nonetheless the paper does convey they underlying process and utility of the appproach, and I am certain that muddy_mine and its web interface will prove invaluable to many. I also look forward to seeing how the project develops further.

I did notice some minor grammatical issues that I recommend you address for the final accepted proof:
Line 31-33: "Over the years, MVs research has been mainly focusing on anaerobic oxidation of methane (AOM)
and the implicit interaction between sulfate-reducing bacteria and methane oxidizing archaea (ANME)
(Bose et al., 2013; Cui et al., 2014)."

Line 34-35: "A myriad of MV studies discussed the thermogenic and biogenic origin of the methane"
suggest:
"A myriad of MV studies discussed the thermogenic and biogenic origin of evolved methane"
(I mean evolved in the chemical sense here, of course!)

Suggest revising to:
"Over the years, MV research has mainly focused on anaerobic oxidation of methane (AOM)
and the implicit interaction between sulfate-reducing bacteria and methane oxidizing archaea (ANME)
(Bose et al., 2013; Cui et al., 2014)."

line 58-60: "To collect all the available data regarding the biological aspects of MVs, we had to exclusively rely on open-access articles. This overreliance was mainly caused by both copyright concerns and the lack of
standardized XML encoding of open-access literature."

Revise to:
"To collect all the available data regarding the biological aspects of MVs, we exclusively rely on open-access articles. This minimises potential issues arising from both copyright concerns and the lack of
standardized XML encoding of open-access literature."


Line 106-107
"Cumulatively, the conveniently centralized taxonomic and physicochemical data can facilitate routine MV-related documentation."

suggest:

"Together, these conveniently centralized taxonomic and physicochemical data can facilitate routine MV-related documentation."

Line 124-125:
"Biomedical fields have always represented the cutting-edge subset of natural science, which actively
implement CS techniques, and are tightly intertwined with the big data term (Luo et al., 2016)."

delete 'the' and 'term':
"Biomedical fields have always represented the cutting-edge subset of natural science, which actively
implement CS techniques, and are tightly intertwined with big data (Luo et al., 2016)."

line 137-138:
"The muddy mine pipeline, however, employs general NLP tools, and so could be used by researchers to mine their data of interest"

Suggest inserting 'own':
"The muddy mine pipeline, however, employs general NLP tools, and so could be used by researchers to mine their own data of interest"